# Effectiveness of a Case Management Intervention Combined with Physical Exercise Compared to Physical Exercise Alone in Older People with High Risk of Falls: A Protocol Study of a Randomized Clinical Trial

**DOI:** 10.3390/healthcare13151814

**Published:** 2025-07-25

**Authors:** Daiene Morais, Karina Gramani-Say, Mariana Luiz de Melo, Ana Laura Oliveira Dias, Verena Vassimon-Barroso, Jean Roberto Ponciano, Daniela Godoi-Jacomassi, Juliana Hotta Ansai

**Affiliations:** 1Department of Gerontology, Federal University of São Carlos, São Carlos 13565-905, Brazil; daienemorais@ufscar.br (D.M.); gramanisay@ufscar.br (K.G.-S.);; 2Graduate Program of Gerontology, Department of Gerontology, Federal University of São Carlos, São Carlos 13565-905, Brazil; marianamelo@estudante.ufscar.br (M.L.d.M.); ana.oliveira@estudante.ufscar.br (A.L.O.D.); danielagodoij@ufscar.br (D.G.-J.); 3Institute of Mathematics and Computer Science, São Paulo University, São Carlos 13566-590, Brazil; jeanponciano@icmc.usp.br; 4Dinâmica—Motor Behavior Laboratory, Department of Physical Education, Federal University of São Carlos, São Carlos 13565-905, Brazil

**Keywords:** accidental falls, aged, case management, clinical trial protocol, health plan implementation

## Abstract

**Background/Objectives**: There is a need for randomized clinical trials with higher quality, especially for older people at high risk of falls, with interventions that consider individual needs, comprehensiveness of care, and connection with primary health care. We designed a randomized controlled trial to examine the effects of a case management intervention combined with a physical exercise protocol on risk factors for falls, falls data, adherence, satisfaction, costs, and implementation in community-dwelling older adults with high risk of falls. **Methods**: A minimum of 60 community-dwelling older people with high falls risk will participate in the randomized controlled assessor-blinded trial (MAGIC—v. 2). The trial will be conducted in a regional health department of São Paulo state (Brazil), which includes 6 cities. Participants will be randomized to the Intervention Group (case management intervention based on all individual risk factors for falls identified by a multidimensional assessment, over 16 weeks, once a week, by telephone calls). Both groups will perform a physical exercise protocol based on falls prevention for 16 weeks (twice a week) in Health Units. The assessment will be performed at baseline, after 16 weeks of intervention, after 6-month follow-up, and after 12-month follow-up. Primary outcome measures include falls data and potentially modifiable risk factors for falls. **Discussion**: This study has the potential to facilitate the future implementation of the intervention based on case management with a focus on fall prevention in the health sectors. **Trial registration**: Brazilian Registry of Clinical Trials (ReBEC).

## 1. Introduction

Approximately 30% of community-dwelling older people suffer at least one fall yearly [1]. In addition to the high prevalence, falls in older people can lead to several serious consequences, such as fear of falling, functional decline, cognitive decline, social isolation, risk of depression, fracture, head injury, institutionalization, and death [2]. According to the Global Recommendation for the Prevention and Management of Falls in Older Adults [3], older adults at high risk of falls have a history of at least one fall in the last year and one or more of the following characteristics: (i) injurious fall, (ii) multiple falls (≥2 falls) in the last year, (iii) presence of frailty syndrome, (iv) inability to get up after a fall without help for at least one hour, and (v) fall accompanied by (suspected) transient loss of consciousness.

Falls constitute a global public health problem with great social impact. Falls prevention is essential, especially among older people with high risk of falls [3,4]. Older people in a more fragile situation and at high risk of falls benefit from individual interventions with a multifactorial approach. One of the multifactorial interventions for reducing falls risk is based on case management [3,5]. Case management intervention consists of identifying problems and conditions that can be changed to meet the identified needs of the older person and family. This care model can offer physical, psychological, and social assistance, which goes beyond traditional health care models [6].

Case management can be an intervention that develops and implements individualized and personalized plans to prevent future falls and reduce their risks and consequences [5,6,7]. In a systematic review, only 12 clinical trial studies involving case management based on falls prevention and reducing risk factors for falls in older people were found until 2023 [5]. There was limited evidence on falls rates and some risk factors for falls. Furthermore, adherence to recommendations varied widely between studies (25 to 88%) and no study assessed satisfaction with the intervention and other factors related to implementation. There is a need for higher-quality randomized clinical trials, especially for older people at high risk of falls, with interventions that consider individual needs, comprehensiveness of care, and connection with primary health care [5].

The Multidisciplinary and Assistance Program of Falls Management in Faller Older People (MAGIC, v. 1) [8] ended in 2024. Despite the positive results of the previous protocol, the present protocol (MAGIC, v. 2) represents an advance in certain respects, including (i) proximity to the public network in order to favor the implementation of the intervention based on case management focused on preventing falls [3] in the context of health care, especially primary care; (ii) target audience of older people with high falls risk, according to the last Worldwide Guideline [3]; (iii) assessments at home instead of remote format; (iv) possibility of investigating minimal clinically relevant differences for the tests used in this population [9,10]; (iv) adding all potentially modifiable risk factors for falls to the case management intervention, since the probability of falls occurring increases as risk factors accumulate [3]; (v) the possibility of using technologies in the intervention; (vi) understanding adherence and satisfaction in a long term; and (vii) having a Control Group submitted to a physical exercise program based on falls prevention in the primary care.

We expect that the case management program combined with physical exercise will show additional benefits compared to physical exercise alone in community-dwelling older people with a high risk of falls. The purpose of this protocol is to evaluate the effectiveness of a case management intervention combined with physical exercise compared to physical exercise alone in reducing risk factors for falls, preventing falls, improving adherence and satisfaction, and assessing costs and implementation feasibility in older people at high risk of falls.

## 2. Materials and Methods

### 2.1. Trial Design

The MAGIC (v. 2) trial is a randomized, controlled, assessor-blinded, single-center (one main research center with collaborators from other centers, with an expectation to cover at least 8 different health territories) superiority trial with two parallel groups. The randomization will be performed as block randomization (random sizes) with a 1:1 allocation (Figure 1).

### 2.2. Study Setting

This research was approved by the Research Ethics Committee of the Federal University of São Carlos (CAAE: 86710225.5.0000.550) on 28 April 2025, and was registered and approved in the Brazilian Registry of Clinical Trials (ReBEC) on 30 June 2025 (code: RBR-8srwcqg). The health system of São Paulo state (Brazil) is divided into 17 regional health departments (DRSs). The DRSs are responsible for coordinating the activities of the State Health Department at the regional level and promoting intersectoral collaboration with municipalities and civil society organizations. This trial will be conducted in the Araraquara DRS (Coração region), which includes six cities (Descalvado, Dourado, Ibaté, Porto Ferreira, Ribeirão Bonito, and São Carlos).

The responsibilities of the Araraquara DRS include the coordination of health actions to ensure the implementation of policies under the Unified Health System (SUS); regulation of access to consultations, exams, and hospital procedures at the primary, secondary, and tertiary care levels, ensuring patients are directed to appropriate services within the region; promotion of formative spaces for the integration and training of professionals to address local and regional demands; and management of human and financial resources in accordance with local needs.

The region was chosen for operational purposes, taking into account that the research center is situated in São Carlos city, which is located in the center and is the reference city of this DRS. According to the latest census conducted in 2022 by the Brazilian Institute of Geography and Statistics (IBGE), the region comprising these six municipalities has a population of 390,525 inhabitants, of which 64,148 (16.54%) are older individuals [11].

### 2.3. Recruitment and Participants

Screening will continue until the target population is achieved. The subjects will be recruited by telephone contact during business hours among eligible individuals and individuals treated at primary to tertiary health units, social and educational services, and from the research group’s database and some community groups. In addition, there will be dissemination through pamphlets, posters, radio and television channels, and social networks, with the assistance of the UFSCar’s communication team. The enrollment period will extend over 9 months.

The volunteers will provide written, informed consent before any study procedures occur. People eligible for the trial will be aged ≥60 years old, resident in the Araraquara DRS (Coração region), non-institutionalized, with a history of at least one fall in the last year, and possibility of contact by telephone. Inclusion criteria will be a level of consciousness that allows the volunteer to be an active subject in the intervention; high falls risk according to the Worldwide Guideline for falls prevention and management for older adults [3], i.e., older adults with a history of at least 1 fall in the last year and one or more of the following characteristics: (i) injurious fall, (ii) multiple falls (≥2 falls) in the last year, (iii) presence of frailty syndrome, (iv) inability to get up after a fall without help for at least one hour, and (v) fall accompanied by (suspected) transient loss of consciousness; ability to walk alone with or without a walking aid; and willingness to participate in the proposed assessments and interventions.

Exclusion criteria will be as follows: a severe and uncorrected hearing and visual impairment that hinders communication during the intervention; neurological diseases that rapidly modify falls risk, including multiple sclerosis, diagnosis of moderate to advanced dementia, epilepsy, and traumatic brain injury, or taking associated medications; and contraindication to physical exercise (acute infectious disease, dissecting aortic aneurysm, severe aortic stenosis, congestive heart failure/unstable angina, acute myocardial infarction, acute myocarditis, acute pulmonary or systemic embolism, thrombophlebitis, ventricular tachycardia, and other dangerous arrhythmias) (Figure 2).

The sample size was calculated based on the primary outcome measures (potentially modifiable risk factors for falls, especially motor risk factors). The sample size was calculated using the statistical program G*Power 3.1, assuming (1) the purpose of identifying the effectiveness of risk factors after 16 weeks of intervention; (2) the test used (two-way ANOVA, considering group and time effects, with Bonferroni test to control type I error due to multiple primary outcomes); (3) the main measure; (4) type I error at 5%; (5) statistical power at 80%; and (6) an effect size of 0.25, a minimum of 28 people should constitute the total sample. Furthermore, to verify the sociodemographic and clinical factors that influence adherence and satisfaction, the rule of at least 10 cases of the outcome (success or failure, depending on which is rarer) was used for each independent variable used in the logistic/linear regression model [12]. Thus, considering an adherence of 80% based on a previous study by the research group [9] and the use of univariate regression analyses, a minimum of 50 people should constitute the total sample. With the chance of losing 20% of participants, a sample of 60 people in total is estimated.

### 2.4. Interventions

#### 2.4.1. Control Group

The Control Group and the Intervention Group will perform the physical exercise protocol for 16 weeks (twice a week) in Health Units, in partnership with trained physiotherapists and physical educators from primary health care and the assistance of the research team. The program will consist of a 10 min warm-up, 20 min balance and gait physical exercises, including how to get up after a fall, 20 min muscle strength physical exercises for the trunk and lower limbs with a functional focus, and 10 min cool-down (muscle stretching and breathing physical exercises), totaling 60 min per session and 120 min per week [13,14]. The physical exercises will have moderate intensity and individualized progression every 3 weeks. Moreover, the participants from the Control Group will be encouraged to maintain their routine and will receive monthly telephone calls for general health guidance.

#### 2.4.2. Intervention Group

Besides the physical exercise protocol, the Intervention Group will be submitted to a case management intervention for 16 weeks, once a week, consisting of the following:

First week: Multidimensional assessment with all potentially modifiable falls risk [3] at the volunteer’s home, at a previously scheduled time;

First–second weeks: Explanation of the risks identified by the case managers to the older person and their families, when appropriate, preferably by video call through the platform of the person’s choice (Google Meet or WhatsApp video) or by telephone at a previously scheduled time. If the volunteer agrees to make the video calls, a video tutorial and PDF will be sent, if necessary;

Second–third weeks: Suggestion of interventions based on the identified risk factors, motivational interview, and individual preferences. The risk factors with the highest intervention priorities will be listed together with the person, preferably by video call or by telephone at a previously scheduled time;

Second–third weeks: Creation of an individualized falls intervention plan, prepared together with the older person, considering treatments for priority falls risks, personal preferences, and available resources, to be carried out preferably by video call or by telephone at a previously scheduled time;

Third–sixteenth weeks: Implementation of the plan by the volunteer, case manager, and other care providers, such as guidance and recommendations, communications with providers, and referrals to specialized programs (through knowledge/understanding of the person’s care network and other community support resources, seeking contacts and information about referrals to programs and subsequent sending of letters, e-mails and/or phone calls with information about the case, in agreement with the person, the family and the health professional/service responsible). If there is a delay in care at the reference service, other strategies will be offered for the identified risk factor, in addition to health education about it and facilitation of communication and integration of services. Monitoring of the plan and its revision, if necessary. Monitoring will be carried out by video calls or phone calls. At this stage, it will be determined whether the person is having difficulty implementing any action, what assistance could be provided, and what modifications could be made (Figure 3) [8].

All cases will be discussed weekly with case managers (undergraduate students in Gerontology at UFSCar and/or gerontologists) and two professionals who are specialists in the area. The case manager must act as an advocate for the older person, their family, and their social support network, in order to defend their interests, empower self-care, autonomy, and independence, facilitate communication with service providers, and coordinate care throughout the network. In this sense, the case managers will receive remote training over 6 months, with real and fictitious case studies.

Furthermore, to facilitate the future implementation of the intervention based on case management with a focus on fall prevention in the health sectors, a partnership was established with the Institute of Mathematics and Computer Science of the University of São Paulo, São Carlos campus (ICMC-USP), Brazil, to create a free Web-based and interactive software for case management interventionists to optimize the management and support their decision-making processes throughout the intervention. The first model was revised in June 2025, after weekly meetings with the creators. The second model will be created by the end of 2025. We expect to have a third and final model in 2026. After that, our research group and the health units will have the opportunity to use the model for case management. We intend to collect data on satisfaction and usability among health professionals that work with falls prevention and health managers.

Thus, we seek to implement strategies for primary health care by checking health data, providing feedback to the older person and their family members, and reporting to the health unit (Secretariat and Ministry of Health) all information regarding the patient. Also, we intend to promote recommendations to achieve long-term adherence and health education, and to investigate the feasibility and applicability of using the software for case management and investigate the costs of implementing the program.

After 16 weeks of intervention, participants in both groups will receive a home physical exercise booklet and will be instructed to follow it until the end of 12 months of follow-up. In addition, participants in the Intervention Group will receive a guidance manual on how to manage the main risk factors for falls and will be guided to maintain the agreed recommendations in case management.

### 2.5. Outcome Measures

Participants will be instructed to wear comfortable clothing and appropriate shoes and use hearing and/or visual aids when necessary. The assessment will be performed at baseline, after 16 weeks of intervention, after 6-month follow-up, and after 12-month follow-up. The evaluators will be blinded and properly trained. All tests will be explained in a clear, simple, and objective manner to the participants. The assessments will be performed at the participants’ homes, if possible, in an environment with minimal noise and distractions.

### 2.6. Primary Outcome Measures

Primary outcome measures include falls data and potentially modifiable risk factors for falls. A fall will be defined to the volunteer as “an event that results in a person inadvertently coming to rest on the ground or other level below and it is not a consequence of a violent blow, loss of consciousness, sudden onset of paralysis or epileptic seizure” [15]. The number of falls and their consequences (lesion, fracture, death, fear of falling, functional difficulty, expenditure, and others) will be monitored in both groups for 12 months through monthly telephone calls, with third party verification to reduce recall bias. Table 1 presents assessments of risk factors for falls.

### 2.7. Secondary Outcome Measures

Clinical and sociodemographic data will be collected at baseline, including age, sex, race, marital status, hospitalization in the last year, years of education, income, level of physical activity (International Physical Activity Questionnaire), and general health (Health Self-Assessment). Furthermore, to assess the minimal clinically important difference (MCID) of the falls risk tools, the Global Change Rating Scale (GRCS) will be used. Thus, after 16 weeks, subjects will answer the following question: “Considering that in the first assessment you were at ‘zero’, use the scale (−7 to +7) to indicate if you are at the same point, better or worse than in the first assessment, considering your risk of falling” [46,47].

Adherence to interventions will be assessed by the weekly frequency of the interventions. Satisfactory adherence will be considered to be equal to or above 70% frequency. A questionnaire will also be used after 16 weeks to assess the reasons for adherence to the interventions (facilitators) and the reasons for non-adherence to the interventions (barriers) [9,48]. In addition, the prevalence and adherence of each recommendation given will be assessed in the Intervention Group (3-point scale: completed as recommended, partially completed, not completed) [9] after 16 weeks of intervention and 6-month and 12-month follow-up.

Satisfactory adherence was defined as a participation rate of ≥70% of the intervention sessions, consistent with previous studies in older populations, which indicate that adherence rates above this threshold are associated with clinically meaningful benefits [9,48]. The target protocol adherence rate of 80% was chosen based on feasibility studies and pilot data from our research group [9], which showed this level of adherence is achievable in community-dwelling older adults engaged in similar interventions.

Furthermore, after 16 weeks, a questionnaire based on the Short Assessment of Patient Satisfaction (SAPS) [49] will be used for both groups. Each item has a scale between 0 and 4 points, and the final score range can vary from 0 (extremely dissatisfied) to 28 (extremely satisfied). In addition, a question about the patient’s overall satisfaction with care will be added (“How would you rate the health care you received in the last 16 weeks?”). Scores range from 0 (worst possible) to 10 (best possible) [9,50].

Finally, we will include the economic variable in the analysis, which will be measured by the incremental cost-effectiveness ratio. It will be calculated based on the difference between the total cost of the IG and the GC. In addition to this variable, we will also analyze the use of health services, measured based on the results of the EuroQol-5D. With the aim of comparing the effects in five dimensions, e.g., mobility, self-care, usual activities, pain/discomfort, and anxiety/depression, obtained in interventions and with scores ranging from 1 to 3 for each item, this instrument is widely used in cost-effectiveness studies [39].

### 2.8. Randomization and Blinding

Participants will be randomly assigned to the Intervention or Control group with a 1:1 allocation using computer-generated randomization software with blocks of random sizes (2, 4, 6). All allocation concealment mechanisms will be conducted by other independent researchers. An independent researcher will open sequentially numbered, opaque, sealed containers with printed randomization numbers and the corresponding group. The researcher then will give information about treatment allocation to the volunteer and care manager staff.

Assessments will be conducted by trained assessors blind to treatment allocation. Due to the nature of the intervention, neither participants nor care managers can be blinded to allocation. They are strongly oriented not to disclose the allocation status of the participant during assessments. The blinded researchers will not have access to information about the allocation until the assessment ends (Figure 4).

### 2.9. Statistical Analysis

A descriptive analysis will be performed. A significance level of α = 0.05 will be adopted for all analyses and the SPSS (22.0) software will be used to perform the statistical tests. The analysis will be performed by intention to treat. The Kolmogorov–Smirnov normality test will be applied to all continuous variables to verify data distribution. To compare groups regarding clinical and sociodemographic characteristics, the chi-square association test for categorical variables, the independent *t*-test for continuous variables with normal distribution, and the Mann–Whitney test for continuous variables with non-normal distribution will be used.

If the normality hypothesis is rejected, the z-score calculation will be performed to standardize continuous data. Volunteers will be classified as having no clinical improvement when the GRCS score is between −7 and 0, moderate improvement between 1 and 3, and great improvement between 4 and 7 after the intervention. The anchoring technique will be used to estimate the minimum clinically relevant difference, using the GRCS as an anchor. Correlations between the GRCS and performance changes of each test will be conducted. The Receiver Operating Characteristic (ROC) analysis will be used to compare the values of each test between volunteers who improved and did not improve according to the GRCS. Through this method, the cutoff score, specificity, and sensitivity of each test will be analyzed [15].

To verify the effectiveness of the interventions, the chi-square association test will be used for categorical variables. About the use of fNIRS, a pilot study was conducted before data collection, and the fNIRS signal demonstrated good quality across the targeted channels and participants, with minimal signal loss or saturation. This assured us the feasibility and reliability of the measurements. Regarding the data workflow, motion artifact control, and ROI selection, at this stage, the specific fNIRS data analysis is still ongoing, and a preregistered analysis plan has not yet been developed. We intend to finish this step until the end of 2025. Using the fNIRS, the digitizer data will be entered into the NIRS-SPM software package (v. 4), implemented in MATLAB 2017 (Natick, MA, USA). The fNIRS data will be preprocessed using custom MATLAB algorithms [51]. For quantitative variables, the two-way ANOVA test will be used to verify the interaction between groups and evaluations, with Bonferroni test to control type I error due to multiple primary outcomes. If an interaction is identified, simple main effects analyses will be performed, with adjustments for multiple comparisons (Bonferroni). To verify the proportion of individuals who did not suffer any fall episode during the 12-month follow-up (“proportion of people who did not have the first fall” and “proportion of people who did not have the second fall”), the Kaplan–Meier analysis will be used.

To verify adherence and satisfaction with interventions and sociodemographic and clinical factors that influence them, descriptive analyses of quantitative and qualitative data will be used, using the MAXQDA2022 software for coding and categorization. The excerpts of the responses will be separated into segments containing the ID of each volunteer and their corresponding code. After coding, the segments obtained will be organized using Bardin’s content analysis [52]. Furthermore, simple linear regression analysis will be used to verify possible influences of adherence (frequency) and satisfaction (score). Also, descriptive analyses and the chi-square association test will be used.

To address missing data resulting from loss to follow-up, we plan to use multiple imputation methods under the assumption of missing at random and to perform sensitivity analyses using mixed-effects models, which can accommodate unbalanced data and account for intra-individual variability over time. These approaches will help to minimize potential bias associated with missing outcome data.

## 3. Discussion

Older adults’ adherence to interventions declines over time, presenting a challenge in achieving sufficient participation, adherence, and persistence in both routine health and social care services [53,54]. Factors such as psychological, cognitive, behavioral, physical, or organizational aspects can influence adherence to a program [55]. Therefore, we believe that case management providing follow-up visits can improve adherence to the program by encouraging confidence by giving older adults an active role and contributing to the success of the present programs for preventing falls [56,57].

The multicomponent physical exercise program is the recommended strategy for improving most, if not all, of the hallmarks of frailty syndrome, including the incidence of falls, poor balance, reduced muscle strength, and poor gait ability. However, it is important to include gradual increases in the volume, intensity, and complexity of the physical exercises in the individual protocol [58]. Due to the need to individualize the program and monitor risk factors, case management is a fundamental care strategy.

Since we intend to reach a larger sample in this program and use challenging tools and protocols, it will be carried out in a hybrid format. We recognize that we may have difficulties moving volunteers to carry out the assessments and part of the training protocol. This factor can be solved by offering travel arrangements for participants to the research center. Furthermore, because this project involves case management, and that brings the volunteers closer to the work team, this factor can be minimized. A previous study showed good treatment credibility based on case management, as well as good adherence to their commendations [9].

Previous studies have reported positive results of case management interventions in reducing risk factors for falls and improving adherence among older adults [5,6,9,10]. For example, Sossai et al. (2024) demonstrated that a case management strategy significantly reduced the incidence of falls in older adults with a history of recurrent falls [6]. Similarly, Janducci et al. (2023) found high levels of treatment fidelity and satisfaction with case management recommendations in community-dwelling older adults [9]. However, these studies had limited scope regarding long-term follow-up and implementation in primary health care settings. Our protocol advances this field by combining the case management intervention with a structured physical exercise program and assessing its feasibility in the Brazilian public health context. The expected results may provide additional evidence to support the integration of these interventions into routine care for older adults at high risk of falls.

Despite the excellent results obtained in the application of the previous protocol [8], we sought, in this second version, to improve the clinical trial. With regard to proximity to the public network: previous studies have concluded that case management is effective for individuals with complex conditions, improving their clinical outcomes and reducing costs for other health sectors [59,60]. Especially in Brazil, the implementation of case management programs in primary care, the gateway for older people to health services, has become an important care strategy [4].

In this second version, we will apply the protocol to older people at high risk of falls, in accordance with the last Worldwide Guideline [3]. The most recent recommendations will be applied, especially with regard to the creation of a case management protocol centered on the older person, including their perspectives and those of their caregivers and family members, following the classifications of falls, their risk stratification, physical exercises, medications in use, multi-domain and multi-component interventions, and telehealth and smart home systems [3].

Another important improvement will be that the assessments will be in-person and not remote, as in the previous protocol the monitoring took place during the COVID-19 pandemic. Furthermore, we hope to identify the minimum clinically relevant difference in tests that assess neuropsychological risk factors for falls and understand the adherence and long-term satisfaction of older people in the intervention.

Since randomized clinical trials are considered the gold standard for evaluating the effectiveness of interventions due to their ability to minimize bias and establish causal relationships between interventions and outcomes, our methodological choice was based on this fact to provide high-level evidence on the effectiveness of our interventions for preventing falls in older adults. In addition, block randomization ensures a balance between groups, and blinding of evaluators reduces detection bias, making the results more robust. Finally, the fact that we conducted the intervention in the primary health care context will allow us to assess not only the effectiveness of the intervention but also the potential implementation challenges that may arise throughout the project, which is extremely important for future implementation of public health strategies.

This study has some limitations that should be considered. For example, the recruitment strategy may favor volunteers who are more motivated or have better access to health services, introducing a potential selection bias. Furthermore, because the intervention involves multiple locations and coordination between different municipalities, logistical challenges, such as participant retention and intervention fidelity, may pose obstacles throughout the study’s development. To this end, the negotiating meetings with the municipalities enabled a more secure alliance between the partners. Despite these limitations, the rigorous design, evaluator blinding, and justification strengthen the study’s internal validity.

## 4. Conclusions

With the improvements made, the application of the second version of the MAGIC protocol could provide reliable information on the effectiveness of case management interventions, especially whether the addition of case management to physical exercise provides incremental benefits compared to physical exercise alone on risk factors for falls, falls data, adherence, satisfaction, costs and implementation, in older adults at high risk of falls. In addition, the detailed methodology of application of the protocol could serve to support the construction of future studies.

## Figures and Tables

**Figure 1 healthcare-13-01814-f001:**
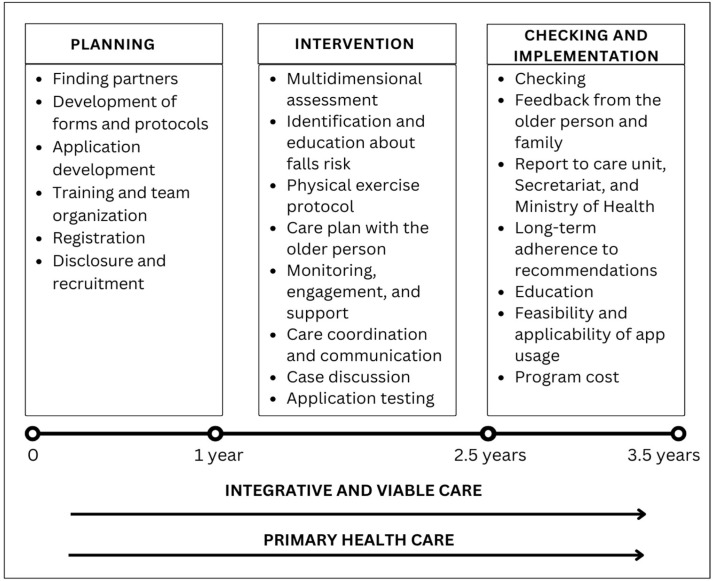
Phases of the MAGIC (v. 2) trial.

**Figure 2 healthcare-13-01814-f002:**
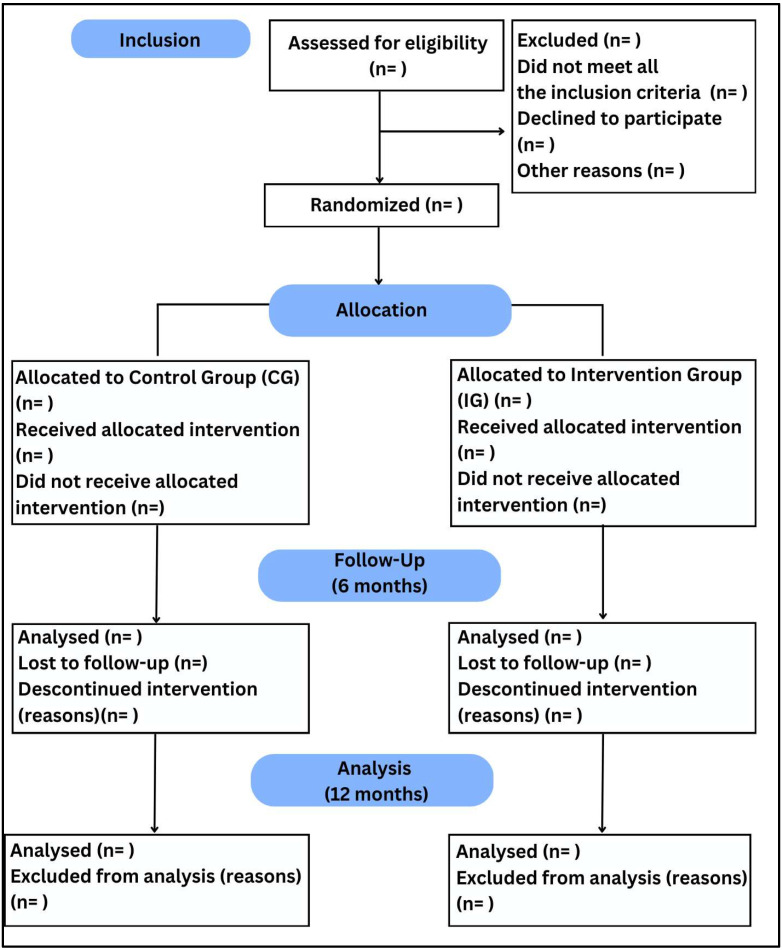
CONSORT flowchart 2010; n = number of individuals.

**Figure 3 healthcare-13-01814-f003:**
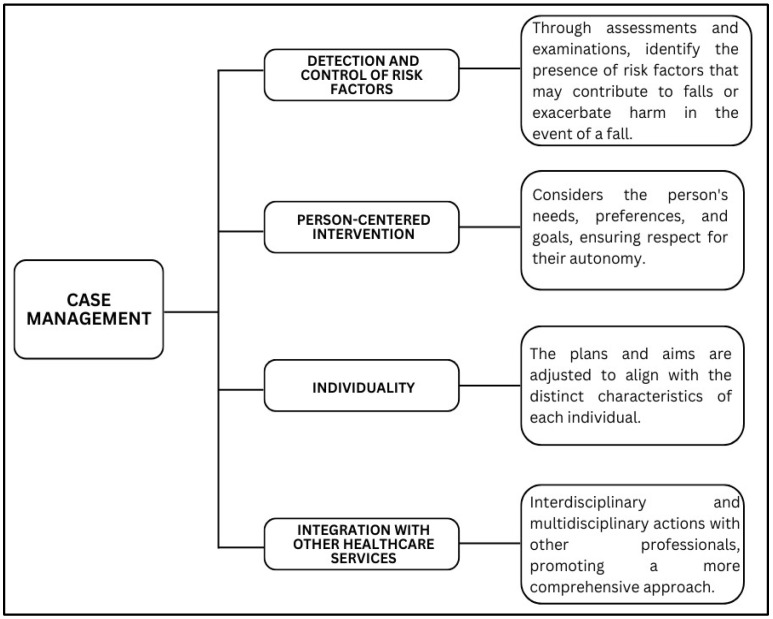
Scheme of the intervention based on case management to prevent falls and reduce risk factors.

**Figure 4 healthcare-13-01814-f004:**
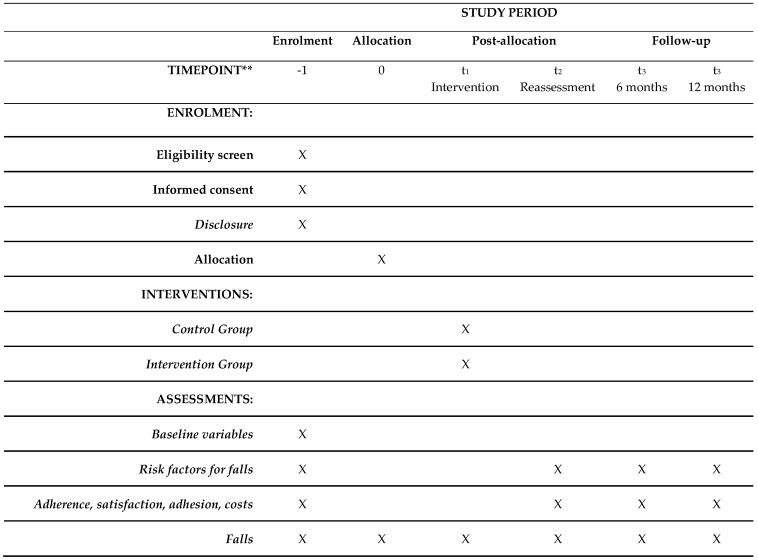
Participant timeline. ** According to SPIRIT checklist, X = task to be performed.

**Table 1 healthcare-13-01814-t001:** Assessment of risk factors for falls.

Risk Factor	Assessment
History of falls [16,17,18]	Number of falls, consequences, circumstances Falls risk score
Feet changes and footwear [19,20]	Michigan Neuropathy Screening Instrument Manchester Index Footwear questionnaire
Cognitive changes [21,22,23,24,25]	Addenbrooke Cognitive Examination (total score, Mini-Mental State Examination, domains scores, Clock Drawing test, fluency verbal test) Brief Cognitive Screening Battery Digit Span Dual task test (Timed up and Go test associated with fluency task, fNIRS) Memory complaint Social Cognition
Depression and anxiety [26,27]	Geriatric Depression Scale Geriatric Anxiety Inventory
Fear of falling [28]	Falls Efficacy Scale—International
Vitamin D deficiency [29]	Blood test
Acute or chronic conditions (osteoporosis, urinary incontinence, cardiovascular disease, dizziness or cerebellar dysfunctions and others) [30]	Presence of morbidities Cardiac frequency Blood pressure
Living alone and other social issues [31]	If they live alone WHODAS 2.0 Social Support Survey
Use of medications [32]	Dose, frequency, and time of each medication, use of psychotropic drugs
Nutritional deficiency [33]	Body mass index Nutritional condition Blood test (calcium, B12 vitamin, K2 vitamin) Mini Nutritional Assessment
Postural hypotension [34]	Symptoms Drop in blood pressure after changing position
Vision and hearing changes [35,36]	Use of bifocal or multifocal glasses Jaeger Eye Chart Difficulty of hearing or people think you have poor hearing Whisper test
Home safety [37]	Home Fast
Use of walking devices and functional activities [38,39]	Use of walking devices Lawton & Brody scale EuroQol-5D Fall to the floor and get up test
Pain [40,41,42]	Brief Pain Inventory Geriatrics Pain Measure
Muscle strength, mobility, gait, and postural balance [43,44,45]	Timed Up and Go test (simple and dual task) Frailty Assessment Short Physical Performance Battery (balance, gait speed, sit-to-stand test) Sarcopenia Assessment Ankle mobility

## Data Availability

The datasets generated and/or analyzed during the current study will be available with the approval of the authors. Anonymized participant-level data will be available following the publication of results by the trial team.

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
