# Peer review of "Effectiveness of a Case Management Intervention Combined with Physical Exercise Compared to Physical Exercise Alone in Older People with High Risk of Falls: A Protocol Study of a Randomized Clinical Trial"

_healthcare, 2025, doi:10.3390/healthcare13151814_

Round 1

Reviewer 1 Report

Comments and Suggestions for Authors

The protocol in the article, although presenting a well-intentioned RCT design comparing case management + exercise with exercise alone in elderly people at high risk of falling, has serious methodological gaps:

First, the sample size was found to be 28 individuals assuming η² = 0.25 (medium effect) with G*Power, then increased to 50 by adding the rule of “≥10 cases per variable” for logistic/linear regression, and completed to 60 individuals with a 20% loss margin; however, it is not stated which primary variable supports this effect size, and adaptations (Bonferroni, Benjamini–Hochberg, etc.) to control type I error due to multiple primary outcomes (fall incidence + more than 26 risk factor tests) are not included, which makes the study questionable in terms of both power and bias.

Although the randomization is blocked, block sizes are not reported, how allocation concealment (e.g. centralized web-based system) will be ensured is not specified, and the single-center design limits external validity; in addition, since exercise was administered to both arms, the assumption of “equal contact time” was violated; weekly one-on-one follow-up in the case management arm may have created an additional behavior-modifying effect (performance/placebo bias); investigators should add a neutral concurrent “health education”-like contact to the control group to offset this difference or include a “treatment time” covariate in the analysis.

Blinding is limited to assessors only; in such behavioral interventions where participant and provider blinding is not possible, it is recommended to include a falls diary and third-party verification to reduce recall bias for the self-reported primary outcome (monthly telephone falls follow-up).

Loss-to-follow-up and “treatment fidelity” were defined as ≥70%, whereas the protocol goal of 80% was not justified; it is unclear whether missing data were handled with multiple imputation or mixed-effects models.

fNIRS measurement is reported only in secondary cognitive subgroups, but data workflow, motion artifact control, and ROI selection are not detailed; pilot signal quality study and preregistered analysis plan should be shared.

Finally, although the multicomponent content of case management (motivational interviewing, telemonitoring, referrals) is well described, cross-referencing the protocol with RAPID-c, TIDieR, or CONSORT-EHEALTH templates, disclosing the source unit prices used in the cost-effectiveness calculation, and specifying user acceptance tests for software support would increase the transparency of the study.

Additionally, the similarity ratio should be reduced.

Author Response

Response to Reviewer 1 Comments

1. Summary

Thank you very much for taking the time to review this manuscript. Please find the detailed responses below and the corresponding revisions/corrections highlighted in the re-submitted files.

2. Questions for General Evaluation

Reviewer’s Evaluation

Response and Revisions

Does the introduction provide sufficient background and include all relevant references?

Can be improved

We have improved all sections based on the reviewer comments.

Is the research design appropriate?

Can be improved

Are the methods adequately described?

Must be improved

Are the results clearly presented?

Can be improved

Are the conclusions supported by the results?

Can be improved

Are all figures and tables clear and well-presented?

Can be improved

3. Point-by-point response to Comments and Suggestions for Authors

Comments 1: The protocol in the article, although presenting a well-intentioned RCT design comparing case management + exercise with exercise alone in elderly people at high risk of falling, has serious methodological gaps: First, the sample size was found to be 28 individuals assuming η² = 0.25 (medium effect) with G*Power, then increased to 50 by adding the rule of “≥10 cases per variable” for logistic/linear regression, and completed to 60 individuals with a 20% loss margin; however, it is not stated which primary variable supports this effect size, and adaptations (Bonferroni, Benjamini–Hochberg, etc.) to control type I error due to multiple primary outcomes (fall incidence + more than 26 risk factor tests) are not included, which makes the study questionable in terms of both power and bias.

Response 1: Thank you for pointing this out. We agree with this comment. Therefore, we have added the primary outcome measures (potentially modifiable risk factors for falls, especially motor risk factors) (pg. 5, paragraph 1). Also, we haved added Bonferroni test to control type I error due to multiple primary outcomes (pg. 5, paragraph 1; pg. 11, paragraph 2).

Comments 2: Although the randomization is blocked, block sizes are not reported, how allocation concealment (e.g. centralized web-based system) will be ensured is not specified, and the single-center design limits external validity; in addition, since exercise was administered to both arms, the assumption of “equal contact time” was violated; weekly one-on-one follow-up in the case management arm may have created an additional behavior-modifying effect (performance/placebo bias); investigators should add a neutral concurrent “health education”-like contact to the control group to offset this difference or include a “treatment time” covariate in the analysis.

Response 2: Agree. We have, accordingly, revised the section 2.8 (pg. 9-10). Actually, we have 1 research main center with collaborators from other centers, with an expectation to achieve at least 8 different health territories (first paragraph, pg. 3). The participants from the Control Group will be encouraged to maintain their routine and will receive monthly telephone calls for general health guidance (section 2.4.1, pg. 7).

Comments 3: Blinding is limited to assessors only; in such behavioral interventions where participant and provider blinding is not possible, it is recommended to include a falls diary and third-party verification to reduce recall bias for the self-reported primary outcome (monthly telephone falls follow-up).

Response 3: Based on previous studies of our research group (Oliveira et al., 2022; Sossai et al., 2025), Brazilian older people do not present good adherence on falls diary because of cultural aspects. However, we have added a third party verification to reduce recall bias (section 2.6, pg. 9).

Comments 4: Loss-to-follow-up and “treatment fidelity” were defined as ≥70%, whereas the protocol goal of 80% was not justified; it is unclear whether missing data were handled with multiple imputation or mixed-effects models.

Response 4: Thank you for your comment. We have clarified the rationale for the 70% threshold for satisfactory adherence and the 80% target adherence rate in the manuscript, which are based on previous studies with older adults and feasibility data from our research group. We have also specified that missing data will be handled through multiple imputations, and sensitivity analyses with mixed-effects models will be conducted to ensure more robust results (section 2.7, pg. 10; section 2.9, pg. 12).

Comments 5: fNIRS measurement is reported only in secondary cognitive subgroups, but data workflow, motion artifact control, and ROI selection are not detailed; pilot signal quality study and preregistered analysis plan should be shared.

Response 5: Thank you for your valuable feedback. A pilot study was conducted before data collection, and the fNIRS signal demonstrated good quality across the targeted channels and participants, with minimal signal loss or saturation. This assured us the feasibility and reliability of the measurements. Regarding the data workflow, motion artifact control, and ROI selection, we acknowledge that these aspects are not fully detailed in the current version of the manuscript. At this stage, the specific fNIRS data analysis is still ongoing, and a preregistered analysis plan has not yet been developed. We intend to finish this step until the end of 2025 (section 2.9, pg. 12).

Comments 6: Finally, although the multicomponent content of case management (motivational interviewing, telemonitoring, referrals) is well described, cross-referencing the protocol with RAPID-c, TIDieR, or CONSORT-EHEALTH templates, disclosing the source unit prices used in the cost-effectiveness calculation, and specifying user acceptance tests for software support would increase the transparency of the study. Additionally, the similarity ratio should be reduced.

Response 6: Although the construction of the software is not the main intervention or outcome of this RCT, we have added more information along the manuscript (abstract, section 2.4.2). The similarity ratio was based on previous studies of falls prevention in older people (Candanedo et al., 2023; Sossai et al., 2024; Reuben et al., 2017; Buto et al., 2019).

4. Response to Comments on the Quality of English Language

Point 1: The English is fine and does not require any improvement.

Response 1: Thank you for your review.

Reviewer 2 Report

Comments and Suggestions for Authors

Dear authors, it is a pleasure for me to have the opportunity to review your manuscript of a trial protocol comparing the effect of case management and physical exercise in older adults at risk of falls.
Please allow me to kindly comment on it:
- Title: - It would be wise to revise its title, as it is very confusing; it is not two interventions being compared, but rather the synergistic effect of combining one (case management) with the other (physical exercise).
- Abstract:

-  Reframe the objective of your research: Is it to compare or measure the effect of an intervention? This will also determine the most appropriate type of study design.

  • Your discussion seems more like a conclusion
  • that could be specified in the ReBEC registration number.
  •  
  • - In your introduction, you define your objective differently, which is to determine whether the case management intervention is superior to the physical exercise intervention. In any case, it seems more appropriate to define not whether it is superior or inferior, but rather which intervention could be highly effective in preventing falls, improving adherence, satisfaction, etc.

  • - Methodology:
    In any case, to propose a clinical trial to compare these two interventions, your methodology would be to not use them together in the experimental group, but rather to have the control group do only exercise and the experimental group do case management. This way, you could see if one is more effective than the other, in the way you propose it. How can you claim that the results in the experimental group are not due to the synergistic effect of applying the two interventions together?
    - On what date was it approved by the ethics committee?
    - It is advisable to follow and accompany the document with a SPIRIT checklist for clinical trial protocols.
    - It should specify whether all the assessment instruments in Table 1 will be included in the study, reference them with scientific literature demonstrating their validation in the target population, and specify the main variables that would be obtained with this excessive battery of assessments.
    - Discussion: Consider justifying the methodological design.
    - A section on conclusions that can be drawn from the results is missing.

Author Response

Response to Reviewer 2 Comments

1. Summary

Thank you very much for taking the time to review this manuscript. Please find the detailed responses below and the corresponding revisions/corrections highlighted in the re-submitted files.

2. Questions for General Evaluation

Reviewer’s Evaluation

Response and Revisions

Does the introduction provide sufficient background and include all relevant references?

Must be improved

We have improved all sections based on the reviewer comments.

Is the research design appropriate?

Must be improved

Are the methods adequately described?

Must be improved

Are the results clearly presented?

Must be improved

Are the conclusions supported by the results?

Must be improved

Are all figures and tables clear and well-presented?

Must be improved

3. Point-by-point response to Comments and Suggestions for Authors

Comments 1: Dear authors, it is a pleasure for me to have the opportunity to review your manuscript of a trial protocol comparing the effect of case management and physical exercise in older adults at risk of falls. Please allow me to kindly comment on it: - Title: - It would be wise to revise its title, as it is very confusing; it is not two interventions being compared, but rather the synergistic effect of combining one (case management) with the other (physical exercise).

Response 1: We appreciate and agree with the suggestion. We have reworded the title of the paper, as well as improved its objective throughout the manuscript to better match the evaluating the relative effectiveness of a case management intervention combined with physical exercise compared to physical exercise alone (title; abstract; introduction section, pg. 2).

Comments 2: Abstract: Reframe the objective of your research: Is it to compare or measure the effect of an intervention? This will also determine the most appropriate type of study design. Your discussion seems more like a conclusion that could be specified in the ReBEC registration number.

Response 2: We appreciate and agree with the suggestion. We have reworded the title of the paper, as well as improved its objective throughout the manuscript to better match the evaluating the relative effectiveness of a case management intervention combined with physical exercise compared to physical exercise alone. This adjustment reflects our pragmatic approach and aligns with the study design, which aims to assess whether the addition of case management yields incremental benefits across multiple outcomes, including fall prevention, adherence, satisfaction, and cost-effectiveness (title; abstract; introduction section, pg. 2).

Comments 3: In your introduction, you define your objective differently, which is to determine whether the case management intervention is superior to the physical exercise intervention. In any case, it seems more appropriate to define not whether it is superior or inferior, but rather which intervention could be highly effective in preventing falls, improving adherence, satisfaction, etc.

Response 3: Thank you for this observation. We agree that the initial wording might have suggested an exclusive focus on establishing superiority. To address this, we have revised the objective throughout the manuscript to emphasize evaluating the relative effectiveness of a case management intervention combined with physical exercise compared to physical exercise alone (title; abstract; introduction section, pg. 2).

Comments 4: Methodology: In any case, to propose a clinical trial to compare these two interventions, your methodology would be to not use them together in the experimental group, but rather to have the control group do only exercise and the experimental group do case management. This way, you could see if one is more effective than the other, in the way you propose it. How can you claim that the results in the experimental group are not due to the synergistic effect of applying the two interventions together?

Response 4: We appreciate the thoughtful review and acknowledge the potential synergistic effect between case management and exercise in the intervention group. However, the decision to offer exercise to both groups was made based on current international guidelines that recommend this as a key intervention for preventing falls in older adults. Although it is not possible to isolate the effects of case management without exercise, our intervention portrays a real-world clinical scenario in which exercise is already a standard care intervention and additional strategies need to demonstrate additional benefit.

Comments 5: On what date was it approved by the ethics committee? It is advisable to follow and accompany the document with a SPIRIT checklist for clinical trial protocols.

Response 5: We have added the date of approval by the ethics committee and the date of rebec registration at the beginning of the methods section (pg.3).

Comments 6: It should specify whether all the assessment instruments in Table 1 will be included in the study, reference them with scientific literature demonstrating their validation in the target population, and specify the main variables that would be obtained with this excessive battery of assessments.

Response 6: We appreciate the careful review and agree with the suggestion. We have added in Table 1 the scientific references that support them, demonstrating their validation in the target population and the variables that will be obtained in each instrument are represented in the first column "risk factor".

Comments 7: Discussion: Consider justifying the methodological design. A section on conclusions that can be drawn from the results is missing.

Response 7: We have justified the study design and added a section on conclusions (Pg. 12).

4. Response to Comments on the Quality of English Language

Point 1: The English is fine and does not require any improvement.

Response 1: Thank you for your review.

Round 2

Reviewer 1 Report

Comments and Suggestions for Authors

The authors have responded sufficiently to my comments, the article is ready for publication.

Author Response

Thank you very much for taking the time to review this manuscript and the valuable contributions. 

Reviewer 2 Report

Comments and Suggestions for Authors

I would like to thank the authors for considering our comments, which have significantly improved their manuscript. I would like to kindly make a few comments:
- Conclusions: I recommend not categorically stating possible conclusions based on the findings they may have with their research, but rather always speaking in the conditional tense: "could provide," "could serve to support," or something like "the difference between groups would represent the additional effect attributable to case management."
- Discussion: I recommend comparing the results you might obtain with your research with those conducted by other authors, naming and referencing them, and including the limitations of your study.

Author Response

1. Summary

Thank you very much for taking the time to review this manuscript. Please find the detailed responses below and the corresponding revisions/corrections highlighted in the re-submitted files.

2. Questions for General Evaluation

Reviewer’s Evaluation

Response and Revisions

Are the conclusions supported by the results?

Must be improved

We have improved all sections based on the reviewer comments.

 3. Point-by-point response to Comments and Suggestions for Authors

Comments 1: I would like to thank the authors for considering our comments, which have significantly improved their manuscript. I would like to kindly make a few comments: Conclusions: I recommend not categorically stating possible conclusions based on the findings they may have with their research, but rather always speaking in the conditional tense: "could provide," "could serve to support," or something like "the difference between groups would represent the additional effect attributable to case management."

Response 1: Thank you for pointing this out. We agree with this comment and have changed the tense in the conclusions.

Comments 2: Discussion: I recommend comparing the results you might obtain with your research with those conducted by other authors, naming and referencing them, and including the limitations of your study.

Response 2: Thank you for pointing this out. In response to your recommendation to compare our expected results with those of other authors and to include the limitations of our study, we made the following changes: 1) We added a paragraph in the Discussion section highlighting the findings of relevant studies, which we had already included in the manuscript, on case management interventions for falls prevention [5,6,9,10]. This comparison emphasizes how our protocol builds on previous work and addresses existing gaps, particularly regarding long-term follow-up and implementation in primary care settings [Paragraph 4]. 2) Study Limitations: We included a new paragraph in the Discussion section outlining potential limitations, such as the relatively small sample size, potential selection and performance biases, and logistical challenges associated with multi-site implementation [Last paragraph of the session].